# Treatment pathways traversed by polycystic ovary syndrome (PCOS) patients: A mixed-method study

Ishwarpreet Kaur[1], Vanita Suri[2], Satya Vati Rana[3], Amarjeet Singh[1]*

**1** Department of Community Medicine and School of Public Health, Postgraduate Institute of Medical Education and Research (PGIMER), Chandigarh, India, **2** Department of Obstetrics & Gynecology, Postgraduate Institute of Medical Education and Research (PGIMER), Chandigarh, India, **3** Department of Biochemistry, All India Institute of Medical Sciences (AIIMS), Rishikesh, Uttarakhand, India

\* dramarjeet56@gmail.com

## Abstract

### Background

This study was undertaken to explore the treatment-seeking pathways traversed by women with PCOS and elicit their behavior, experiences and perspectives regarding it.

### Methods

This concurrent mixed-method study was conducted on 18–40 years old women diagnosed with PCOS at the Gynecology outpatient department, PGIMER, Chandigarh, India. Of the 275 women, who were administered a questionnaire to elicit their treatment-seeking behavior, 62 willing participants were subjected to in-depth interviews. Quantitative responses were descriptively analyzed and presented as count, proportion, mean or median. Framework analysis was performed for the qualitative data. The findings of both types of data were triangulated to construct the pathways to treatment traversed by PCOS patients.

### Findings

Many (~45%) respondents had no information regarding PCOS. Only 9.1% received some information from their doctors. Though the internet was the primary source of information for 37.5% of respondents, they expressed dissatisfaction with the quality of information. Multiple health care agencies were consulted by most (85.8%) of the respondents. Allopathy was the preferred choice of treatment. The average delay in initiating the treatment was 3 months. The major reasons for this were ignorance, the concept of 'normality' and 'endurance'. Deviations from the normal self (like irregular-menstruation, obesity, hirsutism, infertility) were the concern that led them to consult a doctor. They were also dissatisfied with the treatment due to a late diagnosis, lack of relief, taboo, side-effects, expenses involved and the need for repeated laboratory tests. Participants' course of treatment was influenced by the interplay of individual, distress, health-system, and social-economic factors.

**Data Availability Statement:** All relevant data are within the paper and its Supporting Information files.

**Funding:** Dr. Ishwarpreet kaur was supported by the Indian Council of Medical Research, New Delhi

(grant number: 3/1/3/JRF-2013/HRD-040(10175), dated 10.09.2013) for doing her Ph.D. She received financial assistance in the form of a junior and senior research fellowship and contingency grant. The funders had no role in study design, data collection and analysis, decision to publish, or preparation of the manuscript.

**Competing interests:** This study was an ancillary part of a larger PhD project entitled "Efficacy of probiotic-based dietary and lifestyle regime in the management of PCOS cases." The authors declare that they do not have any affiliation with or involvement in any organization or entity with any financial interest, or non-financial interest (such as personal or professional relationships, knowledge or beliefs, political, intellectual or religious interests) in the subject matter or materials discussed in this manuscript. This does not alter our adherence to PLOS ONE policies on sharing data and materials.

## Conclusions

Women with PCOS were dissatisfied with the quality of the information and treatment received. There were treatment delays. The patients consulted multiple health agencies, including indigenous therapies, in the hope of relief. The findings provide an empirical basis on points to focus on for building better coping strategies for managing the condition.

## Introduction

Globalization and industrialization have drastically changed the lifestyle of people leading to the emergence of novel health issues. Polycystic ovary syndrome (PCOS) is one such endocrine condition that has seen a sharp rise worldwide in recent decades. In India, the reported prevalence of PCOS among women of the reproductive age group is 3.7–41% [1–3].

A hormonal imbalance is responsible for most of the signs and symptoms of PCOS, e.g., irregular menstrual cycles, obesity, hyperandrogenism, hirsutism, acne, alopecia, hyperinsulinemia, insulin resistance, and reduced fertility [4, 5]. To get relief from these, patients often consult gynecologists, endocrinologists, dermatologists, dieticians, and psychologists, etc. [6]. Many women suffering from PCOS have reported dissatisfaction with the treatment and information provided by these experts [6–8].

Timely management and provision of optimum knowledge about PCOS may help to improve treatment satisfaction and psychological wellbeing in the patients [6, 8–11]. This will facilitate the active involvement of the patients in their treatment. It will also enhance the quality of doctor-patient interactions [12].

Any delay in diagnosis and treatment of PCOS can increase the risk of its long-term consequences with adverse effects on the quality of life (QoL) of the patients [13]. Recent international evidence-based guidelines for assessing and managing polycystic ovary syndrome have also highlighted the need to improve the QoL aspect of PCOS [14]. Usually, the medical perspective dominates the work of researchers. Hence, not much is known about the treatment-seeking behavior of PCOS patients. Very few studies have explored the patients' experiences and perspectives regarding PCOS treatment [10, 13, 15].

Through this study, we aim to explore the treatment-seeking pathways traversed by women with PCOS and elicit their behavior, experiences, and perspectives regarding it. This will help answer the research questions as to how informed are the women with PCOS about their condition and from where they receive the information; what coping strategies they follow, what barriers they face; what are their views and experiences about the quality of care available for them? And, can we synthesize a treatment pathway from their responses?

## Materials and methods

### Study design and settings

A concurrent mixed method triangulation study [16] was conducted in the Gynecology outpatient department (OPD) of Post Graduate Institute of Medical Education and Research (PGIMER), Chandigarh, India, from August 2016 to January 2018.

The PCOS patients visiting the Gynecology OPD were referred to a separate counseling room after diagnosis and prescription. The patients were informed about the study, and willing participants were recruited after checking for inclusion and exclusion criteria (Fig 1).

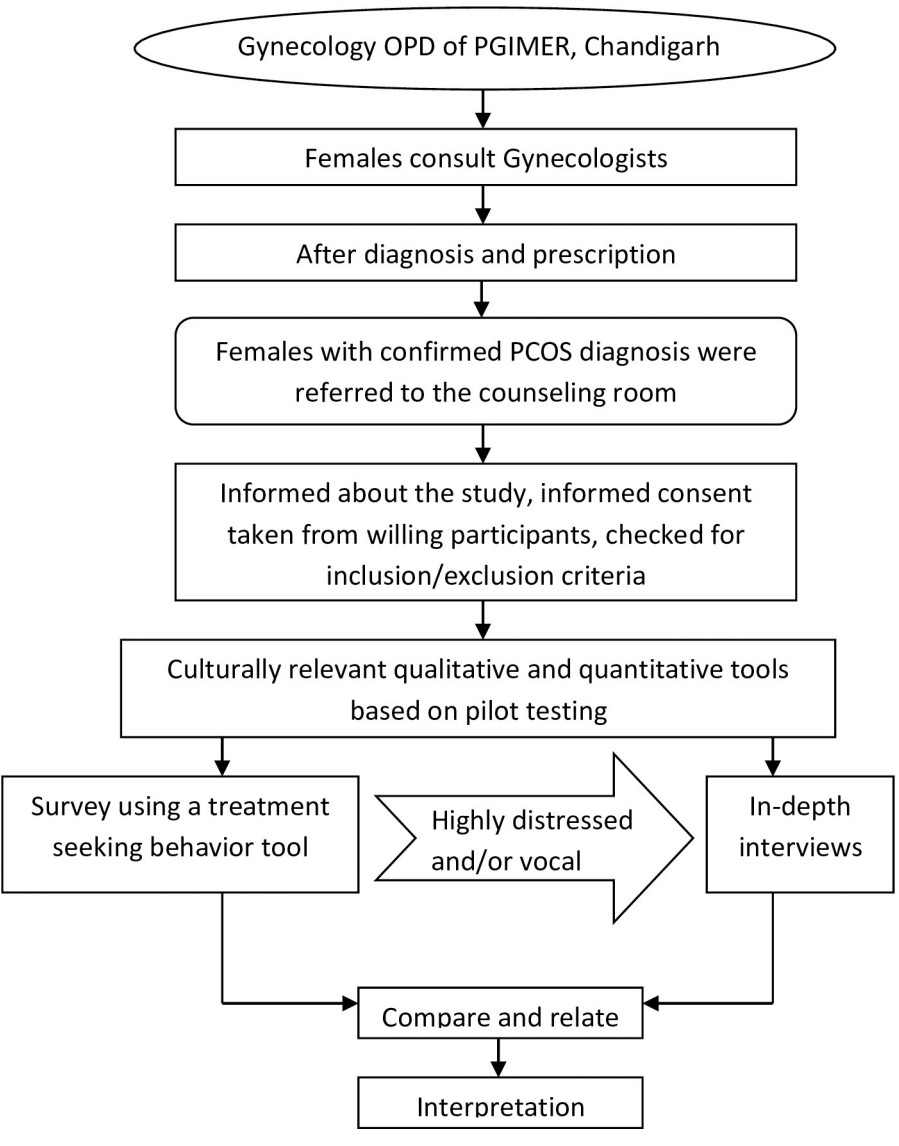

**Fig 1. Study design.**

Family-oriented counseling services on a healthy diet and lifestyle modification were provided to the patients visiting the room, irrespective of their recruitment in the study [17].

Women from different socio-economic statuses and states visit the routine Gynecology OPD for treatment. Out of these, women aged 18–40 years diagnosed with PCOS (Rotterdam criteria) [18] were eligible for inclusion in the study. Women with hyper or hypothyroidism, prolactemia, or any other co-morbidity (like diabetes, cardiovascular disease, pulmonary disease, endometriosis) or pregnancy were excluded from the study.

## Data collection

A treatment-seeking behavior questionnaire was devised and used to gather quantitative information from the subjects on demographics, anthropometric measurements, PCOS awareness status, source of obtaining information, health service utilization rate and switching pattern,

treatment-lag, decision making in seeking health care and choosing health agency, barriers to seeking treatment, opinions on treatment satisfaction and expenditure incurred, etc.

The primary response variable was PCOS awareness status, and the other responses were secondary variables.

For the qualitative data, face-to-face, in-depth interviews were conducted in the presence of the family escorts of respondents. Semi-structured open-ended questions were used to gather relevant data while leaving scope for the generation of additional themes.

The topics covered to understand the coping strategies were 'How was PCOS diagnosed,' 'Reasons for treatment delays,' 'Treatment received,' 'Experiences related to treatment,' 'Choosing apex institute like PGIMER for treatment,' 'Social support system for seeking help,' and 'Understanding of PCOS and experiences of obtaining information.'

The questionnaire was developed after a thorough literature review. It was piloted with eight women before its use in the main study. The interviewer, a Ph.D. research scholar, was trained to conduct qualitative interviews. Consolidated criteria for reporting qualitative research (COREQ) guidelines were followed [19]. The questionnaire was administered in the local language (Hindi), and responses were translated to English. The interviews lasted from 20 to 45 min.

## Sample size

For the quantitative survey, the calculated sample size was 256, which was enhanced to 275 to cover non-response (alpha 0.05 and power 80%, p = 0.20) [6]. The precision (effect size) was taken as 5%.

To study the qualitative aspect, a nested subsample (n = 62) of highly distressed and or vocal respondents were purposively selected, who had the time to spare for an interview with a willingness to share their experiences. The interviews were continued till saturation, i.e., no new codes emerged [20].

This sample is representative of larger PCOS population visiting tertiary care clinical settings.

## Statistical analysis

SPSS 23.0 (IBM, USA) and Microsoft Excel were used for analyzing quantitative data. The categorical data were presented as count and proportions, and continuous data were calculated as mean ± standard deviation (SD) or median (minimum-maximum). Kolmogorov-Smirnov test was done to find the normality of continuous data.

The frequency of Health Care Agencies (HCAs) consulted at each visit gave the HCAs utilization rate. The HCAs consulted by respondents for treatment were categorized as Allopathy government (AG), Allopathy private (AP), and indigenous therapies (IT). Indigenous treatments include Ayurveda, Homeopathy, diet consultation, yoga & meditation.

Sequential switching among the Health Care Agencies (HCAs) was the change in the HCA or the treatment provider from the previous consultation to the next. The respondents who shifted from a particular HCA to another were coded and counted.

For the qualitative data, Excel spreadsheets (Microsoft Excel 2003) was used for data management only. Codes/themes were entered in a spreadsheet after developing them on paper (manually). The interviews were conducted in the local language (Hindi). The subjective responses were noted as field notes and verbatim, transcribed, and later translated into English.

Framework analysis was performed using a conceptual framework, as given in S1 Fig. It covers the key determinants to understand the treatment-seeking behavior for PCOS. The

analysis steps involved familiarization, identification of thematic framework, coding, charting and mapping & interpretation. Using the inductive approach, the codes with similar meanings were grouped under subcategories. Few prior themes were used. New emerging themes were allocated appropriately. The first author and the corresponding author independently read the interviews and regularly met to discuss the interpretations to reach a consensus on the codes and themes [21–24].

For the mixed-method analysis, the results from the qualitative and quantitative findings were integrated by triangulation during the interpretation stage. Both the findings were side-by-side compared and contrasted to identify the patterns, associations, and concepts that agree or converge to build the pathways to treatment traversed by PCOS patients.

Triangulation is a validity procedure where different methods are used to measure the same outcome variable. In our study, the outcomes were analyzed separately, but at the time of interpretation, these were examined for corroboration of results.

### Ethics approval

Approval was granted by PGIMER institute's ethics committee, Ref No. INT/IEC/2015/616, dated 13.10.2015. Informed written consent was obtained from all the participants recruited in the study.

## Results

### Quantitative data results

Overall, 372 participants were contacted. Of them, 59 were not willing to participate, and 38 were not eligible. Participants were mostly young (mean age—24.05 years). The mean weight was 66.78 Kg. Respondents had a mean waist circumference of 94.62 cm. The mean waist to hip ratio was also high (0.92). The average BMI was 27.13 kg/m2. Most (79%) of the participants were overweight. A majority (72%) were under or postgraduates; 47.6% were students, and 65.1% were unmarried. A majority (77.1%) belonged to an upper class or upper-middle class. Respondents were diagnosed with PCOS at an average age of 21.4 (SD 4.7) years. Most (76%) of the study participants were diagnosed between 16–25 years of age. Overall, 112 study participants were diagnosed with PCOS during the preceding year; 14.2% were diagnosed more than five years ago (Table 1).

Many (123; 44.7%) respondents had no information regarding PCOS. For the remaining 152 (55.3%) respondents, the internet was the main source of information in 103 (37.5%) cases. Only 25 (9.1%) study participants received information regarding PCOS from doctors. Newspaper, magazine, and TV were the sources of information for only eight participants. Only sixteen girls obtained information from family/friends.

Of the respondents who delayed treatment-seeking, many (85, 44.7%) considered missing the menstrual cycle as normal; the second main reason for the delay, as told by 75 (39.5%) of the patients, was that they would get some relief whenever they planned to consult a doctor. Few (10; 5.3%) women reported a lack of time, and 4.2% were shy of taking the treatment. Only 5 (2.6%) were scared of consultation. Another 5 (2.6%) said lack of money was the reason for not seeking early treatment.

Two to three health care agencies were consulted by 139 (50.5%) of respondents, while 39 (14.2%) consulted only one agency. Four agencies were consulted by 51 (18.5%) patients and more than four agencies were consulted by 46 (16.7%) of patients.

Fig 2 shows the switching pattern between various HCAs by females with PCOS. Overall, 39 of the 275 respondents did not consult another HCA and came directly to PGIMER. After

**Table 1. Socio-demographic profile, age at PCOS diagnosis and duration of PCOS diagnosis of the respondents (n = 275).**

| Variables | | Mean (SD) |
|---|---|---|
| **Age and anthropometry** | | |
| | Age (years) | 24.1(4.5) |
| | Weight (Kg) | 66.8 (13.0) |
| | Waist Circumference (cm) | 94.6 (12.1) |
| | Waist to hip ratio | 0.9 (0.1) |
| | BMI (Kg/m$^2$) | 27.1 (5.1) |
| **Variables** | | **n (%)** |
| **Education** | | |
| | Postgraduate | 92 (33.5) |
| | Undergraduate | 106 (38.5) |
| | Senior Secondary (11$^{th}$ -12$^{th}$) | 55 (20.0) |
| | Secondary (9$^{th}$ -10$^{th}$) or Upper Primary (6$^{th}$ -8$^{th}$)* | 22 (8.0) |
| **Employment status** | | |
| | Unemployed | 13 (4.7) |
| | Employed | 67 (24.4) |
| | Housewife | 64 (23.3) |
| | Student | 131 (47.6) |
| **Socioeconomic status**# | | |
| | Upper class | 147 (53.5) |
| | Upper middle class | 65 (23.6) |
| | Middle class | 35 (12.7) |
| | Lower middle & below | 28 (10.2) |
| **Religion** | | |
| | Hindu | 163 (59.3) |
| | Others$^\square$ | 112 (40.7) |
| **Marital status** | | |
| | Unmarried | 179 (65.1) |
| | Married˚ | 96 (34.9) |
| **Age at PCOS diagnosis (in yrs)** | | |
| | 10–15 | 22(8.0) |
| | 16–20 | 107(38.9) |
| | 21–25 | 102(37.1) |
| | 26–30 | 30(10.9) |
| | > 31 | 14(5.1) |
| *Years since PCOS diagnosis* | | |
| | <1 | 112(40.7) |
| | 1–2 | 41(14.9) |
| | 2–3 | 37(13.5) |
| | 3–4 | 27(9.8) |
| | 4–5 | 19(6.9) |
| | >5 | 39(14.2) |

* Only 3 went to upper primary school

# BG Prasad socioeconomic classification for 2016. Income (per capita monthly income)

## 23 subjects were in category lower middle class, while 5 subjects were in lower class category

$^\square$ In others, category 5 were Muslims and 1 Christian

˚One subject was married but separated

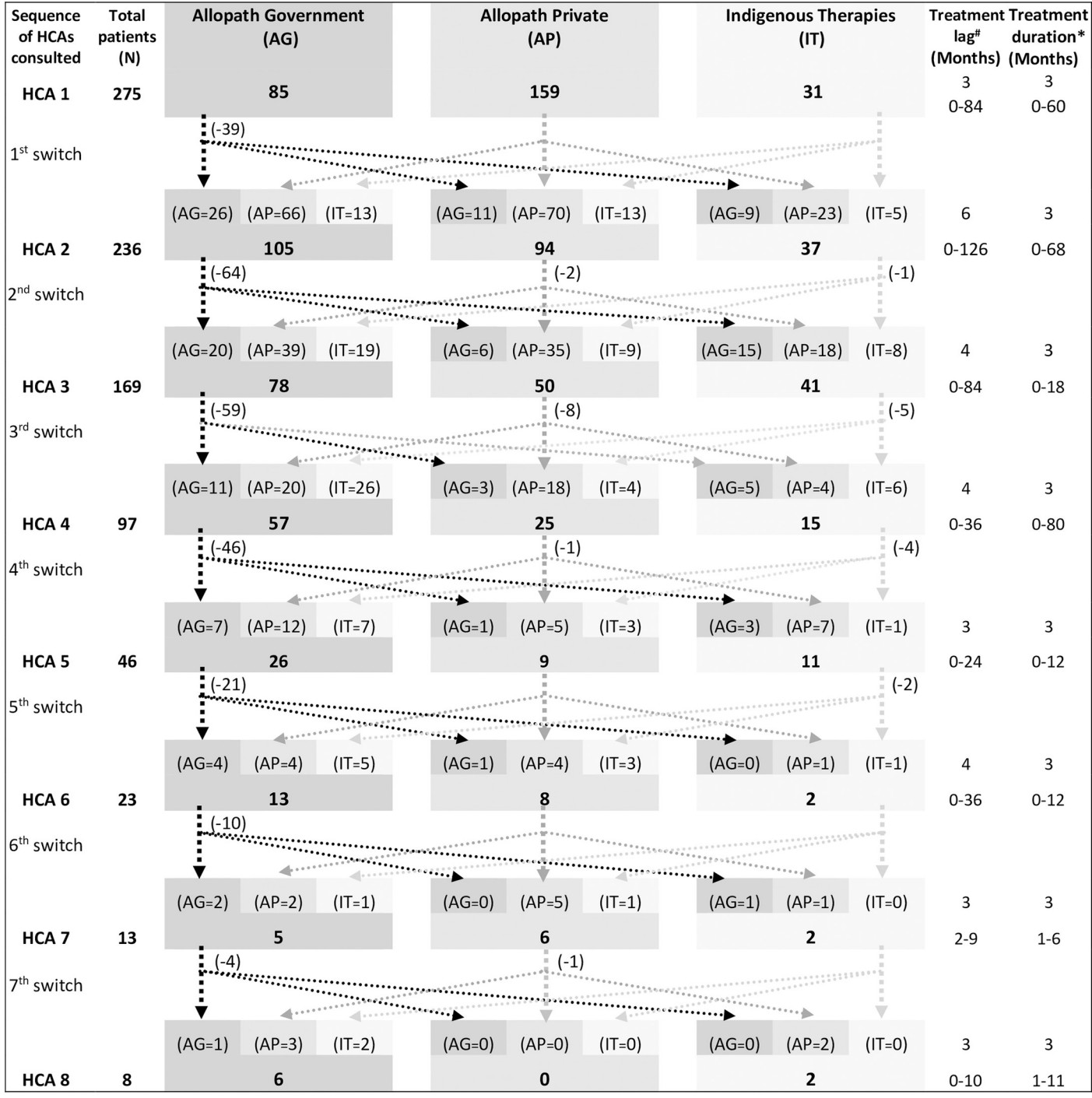

Shift in HCAs from previous HCA is indicated in the parentheses

# Median (min-max) Treatment-lag to approach sequential HCAs for treatment-seeking in months

* Median (min-max) Treatment duration for each HCA consulted

**Fig 2. Pattern of switching between Health Care Agencies (HCAs) by females with PCOS.**

consultation with the first agency, 236 patients went to a second agency. The main shift (66) at this stage was from private Allopathic HCA (AP) to government Allopathic HCA (AG).

Some patients also switched from private and government Allopathic HCA (9+23 = 32) to Indigenous therapies (IT) and vice versa. Similarly, the third HCA was consulted by 169 patients (67 did not continue further). Here also, 39 subjects switched from private Allopathic HCA (AP) to government Allopathic HCA (AG); Some patients switched from private and government Allopathic HCA (15+18 = 33) to Indigenous therapies (IT) and vice versa (19 +9 = 28).

Irrespective of the switching among various HCAs, Allopathy was the preferred choice of treatment, followed by indigenous therapies (Ayurveda and Homeopathy). Few also opted for diet consultation, yoga & meditation. Out of all the HCAs consulted at different time points, the private Allopathy agencies were preferred for the first consultation. Subsequently, public/ government allopathic agencies were selected. The majority of the Ayurvedic, homeopathic, and others health care agencies consulted were private.

The median treatment time lag between the onset of 1st symptom and 1st health care agency consulted was 3 months (range = 1 week to 84 months). The average treatment duration was 3 months. The majority of the respondents (68.0%) consulted first HCA as advised by family or relatives. Many (34.2%) of the respondents found the first treatment to be effective. More than 60% of respondents reported the treatment to be moderate to very expensive.

## Qualitative data results

Themes and codes / sub-themes extracted from treatment-seeking related verbatim responses of the participants are depicted in Table 2.

## Mixed method data results

The experiences and narratives shared by the respondents were collated to come to these findings and help the authors to conceptualize pathways to treatment traversed by PCOS patients (Fig 3).

Five major factors emerged from this data, which seemed to interplay and decided the course of the PCOS treatment pathway followed by the patients, i.e., 1. Individual Factors–Age, education, socioeconomic status, knowledge about disease, symptoms, and compliance. 2. Distress Factors–Perceived sign and symptom severity, peer/family/society reactions to the signs and symptoms, treatment efficacy, fear of treatment. 3. Health System Factors–Screening, referrals, medication, a doctor-patient relationship. 4. Social factors–Family, friends, relatives, and peers and 5. Economic factors–Economic status, treatment, and investigation expenses.

## Discussion

Lack of understanding regarding the nature of PCOS as a disease may lead to poor quality of life (QoL) of patients, often resulting in poor self-image, low self-esteem, anxiety, and depression [8, 9]. Even the delay in diagnosis has been linked to increased anxiety and depression among these patients [6, 10].

The majority of the participants in this study were young, educated, and from the upper and upper-middle classes. Yet, the majority of them were unaware that symptoms of PCOS could be effectively tackled by non-medicinal management. Similar to other studies, half of our study respondents had no information regarding PCOS [6, 25]. Internet was the preferred source of information for 37.5%. However, they had difficulty understanding the information. They Google searched more about diet and weight loss for PCOS. Few of them got worried

**Table 2. Themes and sub-themes of treatment-seeking for PCOS.**

| Title | Theme | Code / Subtheme | Verbatim |
|---|---|---|---|
| Treatment-seeking behavior or coping strategies for PCOS | Barriers to treatment | Concept of normality | "My mother, too, had a menstrual irregularity problem, so I thought it is normal." |
| | | | "I thought it is normal to have irregular menses. Never thought it results due to some underlying disease." |
| | | Ignorance | "I was careless, ignored the problem, considered visiting a doctor as a waste of time." |
| | | Concept of endurance/tolerance | "I would generally get menses once in 2 months; even family did not take it seriously. . . I was balding; probably the weather of the new place did not suit me, or it might be hereditary." |
| | | Relief from symptoms | "My periods would come whenever I decided to visit a doctor after phases of oligomenorrhea." |
| | | Time constraint | "Due to family issues, I was not able to take time out for my health problems. I ignored my health issues." |
| | | Happy to miss periods | "Did not tell my parents that I was having menstrual irregularity because I was happy that they are not coming. I told them after my graduation." |
| | | Concealment of symptoms (Shy/Embarrassed/Fear) | "I was shy and scared of getting an internal check-up by a gynecologist. I was even terrified of injections." |
| | | | "I thought my parents would doubt my character, so I did not inform them about my condition." |
| | | No faith in doctors | "Consulted a doctor after the menstrual cycle skipped; the doctor lied that I am pregnant and later said I had an abortion. None of it was true. My mom-in-law asked me to consult another doctor." |
| | | No escort no consultation | "My husband is too busy; has no time to accompany me to a doctor." |
| | | Poor access to health services | "The doctor was not available whenever I would visit the government hospital." |
| | | Change of Place | "I was taking medicine from *Jalandhar*, but when we shifted to another city, I discontinued the treatment." |
| | | Over the counter medicines / Self remedy | "In between, I would take *ashokarisht* and would get some relief." |
| | Reason for treatment initiation | Menstrual irregularity | "Just 15 days after my engagement, the menstrual problem started; the menstrual cycle was late with spotting issues." |
| | | Bodily changes | "Hair growth was my concern; on doing an Ultrasound investigation, PCOS was detected." |
| | | Fear of infertility | "I got scared when my friend said that an irregular menstrual cycle leads to infertility; I immediately visited the doctor." |
| | | Peer pressure/bullying | "It was when my friends started teasing me that I have kept moustaches like men." |
| | | Incidental Diagnosis | "Got an ultrasound done for kidney stones and came to know about PCOS." |
| | | Friends with a similar problem | "One of our neighbours was taking treatment from PGI for PCOS. She reported improvement, so we also came here." |
| | | False pregnancy alarm | "When the menstrual cycle skipped, we thought it could be due to pregnancy, but it turned out to be PCOS." |
| | Advice by Doctor | New normal | "Doctors said nothing to worry about regarding PCOS, these days 75% of girls have this problem." |
| | | Concept of normality | "Doctors said nothing about it. She is young; her hormones are changing. With time, everything will be normal." |
| | | Doctors ignorant about treatment | "Doctor said, 'Come later after marriage, if unable to conceive'." |
| | | | "PCOS has become very common these days (75% girls have it); so nothing to worry about. " |
| | | Modify Lifestyle | "My cousin is a doctor; he asked me to correct my diet and do yoga." |
| | | Hormone treatment not for adolescents | "I consulted a doctor. But he said, 'You should wait; I cannot start hormones at this age'." |
| | Advice received from family and relatives | Concept of normality | "Relative said there is nothing to worry about; even her daughter had an irregular menstrual problem. Nothing serious about it." |
| | | No cure for the problem | "Friends said, there is no cure for PCOS and in future, I might face problems like infertility." |
| | | Awareness about disease and remedy | "Everyone advised me to lose weight and the problem will resolve. They told, 'Don't go after medicines'." |
| | | Faith in PGIMER | "My uncle was successfully treated here. Our family has faith in PGI; so I also came here." |
| | | Consult Doctor | "When my aunt came to know about my condition she asked my mother to consult a doctor." |
| | | AYUSH / indigenous treatment | "Relatives said allopathic medicine has side effects; so thought of starting homoeopathic treatment." |
| | | Home remedy/concept of food as a cure | "Friends and relatives said,' her cycles will get regular on its own; just give her 'hot' things to eat." |
| | Problem with the treatment | Delayed diagnosis | "It has been more than a month; they are just getting tests done. No medicine is given so far." |
| | | Ineffective | "Treatment is not effective. The only acne has subsided, but menses are still irregular." |
| | | Dissatisfied (Limited options/side-effects/taboo) | "Have consulted infinite no. of doctors, but had no relief. Would get menses only with 21 days 'medicine." |
| | | | "Do not get menstrual cycle without taking medicines, due to drug dependence." |
| | | | "I did not take OCPs prescribed by the doctor; these are for married women." |
| | | Expensive | "Blood tests easily costs Rs 6000/- from private clinics." |
| | | | "Private doctors charge nothing less than Rs 500–800 for a single consultation." |
| | | Repeated investigations | "Every time I consult a new doctor, the tests are repeated; even when I show them previous reports." |
| | | | "I am on medication for 3 years; have gained weight due to treatment." |
| | | Difficult to lose weight | "My sister is a doctor; she asked me to reduce weight; but I was not able to do it, despite my efforts." |
| | Choosing PGIMER for treatment | Expert opinion | "Not taken medicines prescribed by the private doctor; thought of taking 2nd opinion from PGI." |
| | | PGIMER–the best hospital | "A doctor advised me to get ovarian drilling; so I came to PGI for their opinion. PGI is the best." |
| | | On referral by doctor/significant other | "From last 3 months, the menstrual bleeding was not stopping. I came for treatment to PGI, as my sister was taking treatment from here only. She asked me to come along." |
| | | Poor response to AYUSH | "Homeopathic medicines stopped having any effect on my condition; so I came to PGI." |
| | PCOS awareness and quality of information | Ignorance / No information | "I have no information about what PCOS is; and that I am suffering from it." |
| | | Confusion | "What is the difference between PCOS and PCOD? One doctor said I have PCOS and the other PCOD. I am confused." |
| | | Difficult to understand | "Did try to get information about PCOS on the internet, but didn't understand anything." |
| | | Inadequate information provided by doctors | "Doctor told me, 'your ovaries are filled with water." |
| | | | "I am told that I have cysts in ovaries due to hormonal imbalance". During the interview, she asked, 'Does this mean I have cancer?' " |
| | | The authenticity of information on the internet doubtful | "If you search any disease on the internet, the concluding lines almost always state that the disease can lead to cancer and death." |

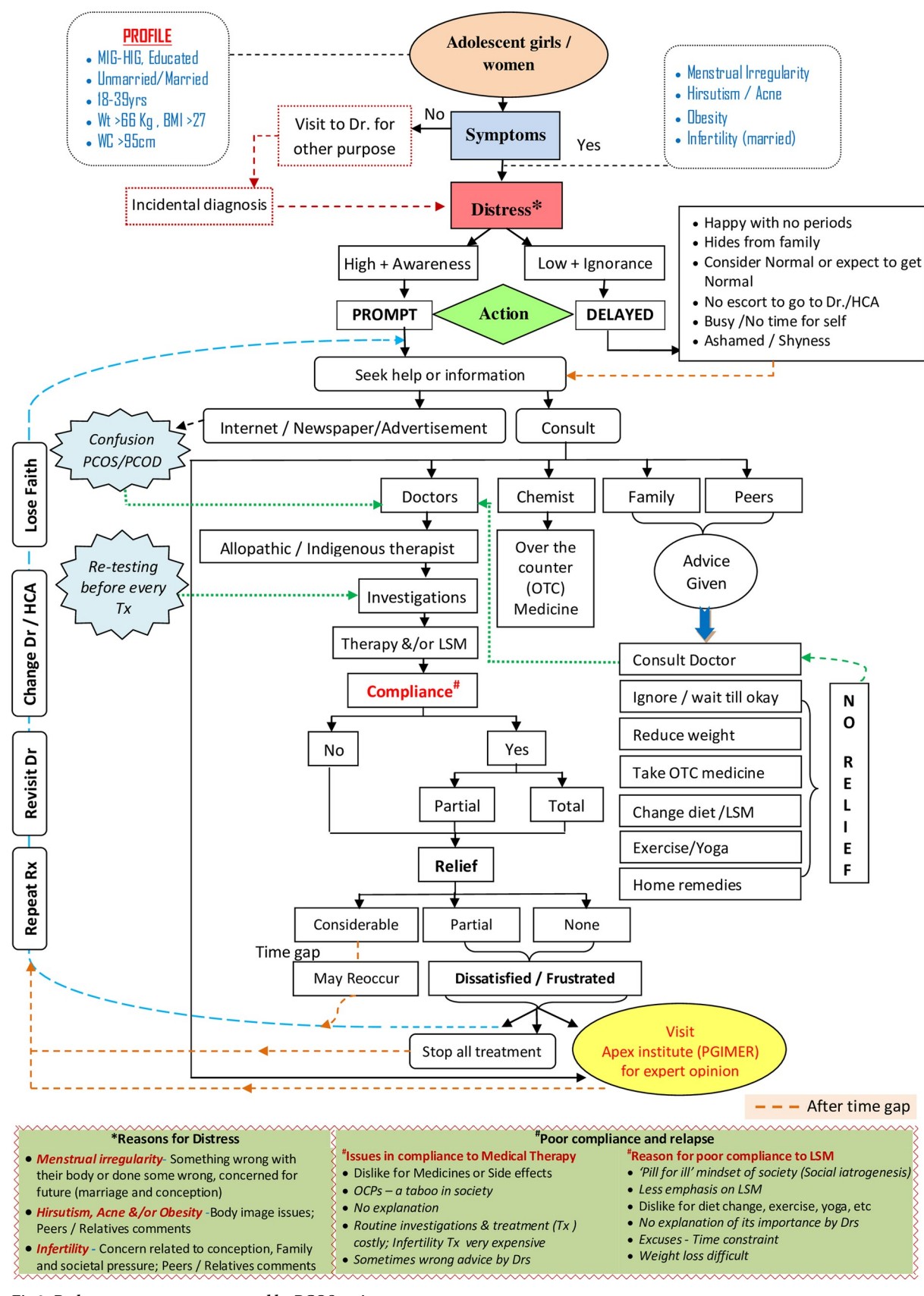

**Fig 3. Pathways to treatment traversed by PCOS patients.**

after reading the information on the internet. A participant said, "I avoid reading about PCOS on the internet. The information gives me stress and tension." The majority of respondents were ignorant that they were suffering from PCOS, despite the diagnosis mentioned on their cards. Only one in ten participants received information regarding PCOS from their doctors. Even that was too difficult to understand. They were scared and hesitant to ask anything from the doctors.

In our study, around 14% of participants were suffering from the condition for more than 5 years, with an average of 3 months' treatment lag. They considered the problem as self-limiting. One of the respondents stated, "I thought that the problem would resolve on its own. The problem might have occurred due to change of place. Since the time I joined the hostel, my periods have become irregular."

The concept of normality was another major reason for such delay. Our respondents stated, "Many girls suffer from irregular menstrual cycles. Therefore, there is nothing to worry about"; "I thought it is normal to have irregular menses. I never thought it a result of some underlying disease." In any society, if a majority of people suffer from similar symptoms, it is considered 'normal' [26]. Even the women suffering from frequent urinary tract infections or uterine prolapse tolerate their condition for a long time before consulting any health agency. They often suffer silently for years, thinking the condition to be 'normal' [27, 28]. Such an attitude perpetuates the concept of suffering in silence in a woman's life. They consider their health problems as an inevitable part of their lives [27].

As per our results, a key barrier to delay in treatment-seeking was the high tolerance threshold of women. It is a common practice among Indian women to consult doctors quite late for their health problems. They tolerate their condition for long without sharing it with anyone until the severity is aggravated [29]. In third world countries, because of modesty or shame associated with reproductive health issues, 'culture of silence' is a key reason for the delay in treatment [30, 31].

Deviations from the normal self (like irregular menstruation, weight gain, acne, hirsutism, infertility) were the concern that led the PCOS patients to consult a doctor. India is a family-centered society, and kinship has a major say in personal and health issues. Many respondents consulted HCAs as advised by their relatives.

Allopathy is usually the choice of treatment in our society [32, 33]. Even in our study, it was the preferred choice of treatment in all the eight agencies consulted. This is in concordance with the prevailing health culture of 'pill fixation' through which people look for quick relief without even bothering to know about the disease etiology [34].

In most cases, patients consulted private health care providers as the first agency. This could be due to their desire to avoid inconvenience and long waiting time in government hospitals [35]. However, later, their preference shifted to government hospitals, when they did not get any relief despite the high treatment cost of private consultation.

Being ignorant about the etiology or remedy of PCOS, they try multiple health care agencies–Allopathic, Homeopathic, Ayurveda, etc. In our study also, around 60% consulted three or more agencies (Allopathy or Indigenous). They were not happy with the repetition of similar treatment and investigations. Patients complained that the health care providers did not explain the 'what and why' of the treatment. "All the doctors I have consulted gave me the same medicine for irregular menses. Earlier, there were cysts only in one ovary. Now, both ovaries have cysts." "Do not like to take hormonal pills. These are for married women." "Menstrual cycle is regular till I take medicines; the moment I leave them, the problem recurs." Most of them were, naturally, dissatisfied with such treatment. They frequently changed treatment agencies in the hope of getting better results. One-third of the respondents in our study reported no relief by the treatment taken from 1st agency. Often doctors also recommended

weight reduction and lifestyle modification to them. However, no one taught them practical tips on how to adopt a healthy lifestyle.

Their dissatisfaction with the treatment is reflected in the verbatim, "Do not get menstrual cycle without taking medicines"; "I have lost count of doctors consulted"; "No point going to doctor, again and again, every time I go, I am given a hormonal tablet"; "Treatment is not effective. Acne has subsided, but menses are still irregular."

The delay in diagnosis and treatment due to a long list of laboratory investigations was an issue for some girls.

There was a lot of confusion regarding which health agency to choose. Few directly consulted apex institutes like PGIMER. Being a tertiary care hospital, patients non-responsive to treatment are referred to PGIMER from north India. It has a good reputation, and people prefer to consult it for their ailments. One patient told, "Consulted my family doctor many times. But, the problem was not cured and recurred. So, he referred me to PGIMER."

PCOS patients are often exploited by quacks, who put them on some indigenous medicine. They end up wasting a lot of money. Quite often, their symptoms and problems keep worsening. They often lose hope of an easy recovery.

In the 21st century, the focus of evidence-based medical care has a strong emphasis on diagnostic tests, which are quite costly. This makes the overall treatment very expensive. If one goes by the PCOS guidelines for diagnosing the condition, the overall cost of investigations will come close to Rs. 5000/- ($68.6) or more if one gets tests done from a private laboratory. The cost increases many folds in the case of PCOS women seeking infertility treatment. In our study also, some of the infertile women had spent more than 3 lakh rupees ($4116.9) on infertility treatment. The annual economic burden for diagnosis and treatment of women with PCOS is reported to be around 4.36 billion dollars in the US [36]. This is evident from patients verbatim "Have no idea how much money has been spent so far on infertility treatment. We are under debt due to such expensive treatment for infertility." "I must have spent 2–3 lakh rupees on infertility treatment."

The treatment pathway illustrated in this study indicates that the level of distress associated with the PCOS symptoms and the lack of adequate information affected the action taken by the patients. Prompt action was associated with a high level of distress and better awareness level, while ignorance and other barriers delayed the treatment. The action involved either seeking help or information from health professionals or other sources like chemists, family, peers, the internet, or other media. Few girls directly came to apex/tertiary care institutes (like PGIMER).

The degree of relief obtained varied as per the degree of adherence to the treatment prescribed. Few patients got considerable relief, while others got frustrated due to partial or no relief. In either case, they repeated the treatment. Some of them revisited the same doctor; others changed the doctor or the HCA. Many stopped the treatment. After some time gap, when the symptoms reappeared or worsened, they again sought help. This cycle continued till they got considerable relief, reached an apex institute, or got exasperated.

## Strengths and limitations

The weaknesses of the study include potential recall bias and selection bias for the qualitative aspect. Also, the respondents may just be considered representative of PCOS patients reporting to gynecology OPD of any tertiary care hospital of India (and not of the general population).

Despite these limitations, this study is the first extensive and comprehensive study from north India that corroborated quantitative and qualitative research techniques to ensure a deep understanding of the treatment pathways traversed.

## Conclusion

This study suggests that women with PCOS faced difficulties while traversing the treatment pathways. Despite taking treatment, they were ignorant about their condition. They were dependent on the internet as their main source of information. There was an average delay of 3 months before seeking treatment. The trigger to initiating treatment was usually distressed over the body and physiological changes. They were not satisfied with the treatment received. PCOS routine treatment was perceived to be expensive and not very effective. Multiple health agencies consultation and frequent switching of therapies were common. Allopathy was the preferred agency of treatment.

Overall, the treatment pathways opted by PCOS patients depends on a combination of factors, e.g., individual distress, health-system related issues, and social-economic circumstances.

## Recommendations

The findings provide an empirical basis on points to focus on for building better coping strategies for managing the condition. Physicians and health care agencies should abide by the treatment guidelines. They should empower PCOS patients for self-care. They should address their concerns (related to disease, therapy, body image, stress, etc.) and individualize the PCOS management goals. Easy to understand information based on recommended guidelines must be provided to them.

## Supporting information

**S1 Checklist. COREQ checklist.**
(DOCX)

**S1 Fig. The conceptual framework of treatment-seeking behavior for PCOS.** It was developed from a literature review of the determinants affecting treatment-seeking behavior.
(TIF)

**S1 File. Dataset analyzed for the study.** It contains both qualitative and quantitative data set along with the variable coding scheme.
(XLSX)

## Acknowledgments

We gratefully acknowledge all the study patients who gave their time and commitment to this study.

## Author Contributions

**Conceptualization:** Ishwarpreet Kaur, Amarjeet Singh.

**Data curation:** Ishwarpreet Kaur.

**Formal analysis:** Ishwarpreet Kaur, Amarjeet Singh.

**Investigation:** Ishwarpreet Kaur, Satya Vati Rana.

**Methodology:** Ishwarpreet Kaur, Vanita Suri, Satya Vati Rana.

**Resources:** Ishwarpreet Kaur, Vanita Suri, Amarjeet Singh.

**Software:** Ishwarpreet Kaur.

**Supervision:** Vanita Suri, Amarjeet Singh.

**Validation:** Amarjeet Singh.

**Visualization:** Vanita Suri, Satya Vati Rana.

**Writing – original draft:** Ishwarpreet Kaur.

**Writing – review & editing:** Vanita Suri, Satya Vati Rana, Amarjeet Singh.

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
