## [Decision Letter · Decision Letter 0]

3 Jun 2021

PONE-D-21-07635

Treatment pathways traversed by polycystic ovary syndrome (PCOS) patients: A mixed method study

PLOS ONE

Dear Dr. Singh,

Thank you for submitting your manuscript to PLOS ONE. After careful consideration, we feel that it has merit but does not fully meet PLOS ONE’s publication criteria as it currently stands. Therefore, we invite you to submit a revised version of the manuscript that addresses the points raised during the review process.

The two reviewers have identified some major concerns in the manuscript that are well described in their comments. Many of the issues were regarding the need for greater brevity and clarification

We look forward to receiving your revised manuscript.

Kind regards,

Stephen L Atkin, MD

Academic Editor

PLOS ONE

Journal Requirements:

2. In your Methods section, please provide additional information about the participant recruitment method and the demographic details of your participants. Please ensure you have provided sufficient details to replicate the analyses such as: a) a statement as to whether your sample can be considered representative of a larger population, e) a description of how participants were recruited, and b) descriptions of where participants were recruited and where the research took place.

Given these details, please consider whether the design of your study is such that the results support your statement: "Majority of the participants in this study were young, well educated and from upper and upper middle class indicating that, by and large, PCOS is a disease of young, literate and ‘well off’ women."

Please note that our publication criteria state that data presented in the manuscript must support the conclusions drawn.

7. Please provide additional details regarding participant consent. In the ethics statement in the Methods and online submission information, please ensure that you have specified what type you obtained (for instance, written or verbal, and if verbal, how it was documented and witnessed). If your study included minors, state whether you obtained consent from parents or guardians. If the need for consent was waived by the ethics committee, please include this information.

Reviewers' comments:

Reviewer's Responses to Questions

**Comments to the Author**

1. Is the manuscript technically sound, and do the data support the conclusions?

Reviewer #1: Partly

Reviewer #2: No

2. Has the statistical analysis been performed appropriately and rigorously? 

Reviewer #1: Yes

Reviewer #2: Yes

3. Have the authors made all data underlying the findings in their manuscript fully available?

Reviewer #1: No

Reviewer #2: No

4. Is the manuscript presented in an intelligible fashion and written in standard English?

Reviewer #1: Yes

Reviewer #2: No

5. Review Comments to the Author

Reviewer #1: This manuscript presents concurrent mixed method study on data generated from women (18-40 years old) diagnosed with PCOS recruited at PGIMER, India to explore the treatment-seeking pathways traversed by women with PCOS, in regards to their behavior, experiences and perspectives. The study was approved by the respective Ethics Board. While the study objectives are exciting and important, my comments are below:

1. The Abstract requires a major facelift. The Methods section simply states "mixed-method" study, without any specific pointers to the respective statistical analysis conducted. The Results section doesn't state the strength and direction of the findings (via estimates and p-values); this may not be appealing to a reader interested in also understanding the strength of the directions, and findings.

2. The sample size/power statement does not specifically mentions a desired effect size, the statistical test used, and whether it was computed "using the primary response variable". In fact, it was hard to understand which one is the primary response, and which are secondary (if any).

3. More details on exactly what methods (and how, like which software) were used under the banner of "mixed methods" is missing in the Statistical Analysis section. For example, how was the switching calculated in Table 2? I understand triangulation was done, but it may not be very familiar to a reader with little exposure to mixed methods. Note, this is not the same as running a logistic regression.

Reviewer #2: Treatment pathways traversed by polycystic ovary syndrome (PCOS) patients: A mixed

method study.

This is an exciting piece of research addressing the awareness, information source and coping strategies in women with PCOS.

Comment: the abstract section of the study is too long. The method section should summarise concisely the method used. Similarly, the result section of the abstract should only summarise the main results. The conclusion should be more focused and reflect the results of the study. Overall, the abstract should be flawless for the readers, not too extended and more focused.

Comment: Financial disclosure, please provide grant number or reference.

In the introduction, the author claimed that “Polycystic ovary syndrome (PCOS), a gynaecological morbidity". PCOS is an endocrine condition affecting women of reproductive age; the primary pathology is hormonal disturbances which drives to metabolic, gynaecological and reproductive consequences.

Comment: in the introduction, the author should refer to the recent "international evidence-based guideline for assessing and managing polycystic ovary syndrome 2018". Which highlighted the importance of evaluating the QoL aspect of PCOS.

Comment: the study was designed as a mixed-method triangulation; this method is suitable when combining quantitative and qualitative methods to answer a specific research question. This method leads to a better explanation and link different aspects of a single research question.

Comment: in the method section, please move reference 16 to Rotterdam criteria instead of the end of the text.

Comment: Data collection: this section should be the intervention section which will reflect the questionnaire administered in the study as the author was used an Excel spreadsheet to collect data. For the quantitative data, it is not clear whether the questionnaire used was a validated questionnaire? What was the scale used to capture responses, for instance, the 7-point-Likert scale?. The author has piloted the questionnaire with eight patients; what were the results? Whether any modification to the questionnaire was made as a result of piloting?

Comment: COREQ guidelines, please use full text (no abbreviation) when it first used.

Comment: The patient's responses were captured in another language which later translated into English. Does this create any risk of bias for the accurateness of the translation? Did the author use approved translation software or translator? It isn't easy to translate word to word from any other language to English.

Comment: The technical details should be expanded and clarified to ensure that readers understand exactly what the researchers studied.

Comment: the statistical analysis session is too long; this should be short and concise.

Comment: the author mentioned, "This study objective was an ancillary part of a larger PhD project entitled “Efficacy of probiotic-based dietary and lifestyle regime in the management of PCOS cases", this should be removed from the method section of the manuscript and should be in the conflict of interest section.

Comment: the manuscript lacks clear inclusion/exclusion criteria for the participants.

Comment: in the quantitative section of the results, please give exact numbers instead of writing “remaining”.

Comment: what is “viz” mentioned in the manuscript?

Comment: table 2 is complicated to follow the trends, very complicated for readers to understand precisely what is happening. It should be simplified further.

Comment: overall, the results section is very long. This should be more concise and focused.

Comment: the results and the discussion should be separate sections.

Comment: the discussion section is very repetitive. Please rewrite this section and make it more concise.

Comment: Line 447, Figure 1: Pathways to treatment traversed by PCOS patients. This should be removed from the discussion and only refer to the figure within the text, for example (Figure 1). Please remove any bullet points from the text. This section should be in the result section.

Comment: the author claimed, “Five major factors emerged from this data, i.e."; this should be in the result section instead of the discussion.

Comment: one of the limitations of this study is that the questionnaire was administered in a language other than English, then translated to English. This could lead to a risk of bias in interpreting the script word to word.

Comment: the conclusion section is very poor; hardly any main results were mentioned. Additionally, this section is very long; it must be short, concise and summarising the main results of the study.

Comment: reference 1, 23, 25, 26,29,32, 33,34, please add DOI.

Comment: overall, the language is unclear, making it difficult to follow. I advise the authors work with a writing coach or copyeditor to improve the flow and readability of the text.

Comment: what this study will add to the field?

Comment: reading this study, again and again, I do not know what precisely the main results are. This should be very clear and consistent throughout the text, from the abstract to the conclusion.

6. PLOS authors have the option to publish the peer review history of their article (what does this mean?). If published, this will include your full peer review and any attached files.

Reviewer #1: No

Reviewer #2: **Yes: **Mohammed Altigani Abdalla

---

## [Author Response · Author response to Decision Letter 0]

13 Jul 2021

Responses to comments made by the academic editor and reviewers are:

• Comment 1: Please ensure that your manuscript meets PLOS ONE's style requirements, including those for file naming.

Response: We have ensured that the manuscript meets the PLOS ONE’s style requirement, including file names.

• Comment 2: a) In your Methods section, please provide additional information about the participant recruitment method and the demographic details of your participants. Please ensure you have provided sufficient details to replicate the analyses such as: a) a statement as to whether your sample can be considered representative of a larger population, e) a description of how participants were recruited, and b) descriptions of where participants were recruited and where the research took place.

b) Given these details, please consider whether the design of your study is such that the results support your statement: "Majority of the participants in this study were young, well educated and from upper and upper middle class indicating that, by and large, PCOS is a disease of young, literate and ‘well off’ women."

c) Please note that our publication criteria state that data presented in the manuscript must support the conclusions drawn.

Response: a) We have provided all the additional information mentioned in the comments in the methods section of the revised manuscript.

b) No, the study design does not favour participants from any particular socio-economic or demographic status. The study was conducted in tertiary care public hospital where people from all strata visit for treatment.

c) We have revised the conclusion section.

• Comment 3: We suggest you thoroughly copyedit your manuscript for language usage, spelling, and grammar. If you do not know anyone who can help you do this, you may wish to consider employing a professional scientific editing service. 

•The name of the colleague or the details of the professional service that edited your manuscript

•A copy of your manuscript showing your changes by either highlighting them or using track changes (uploaded as a *supporting information* file)

•A clean copy of the edited manuscript (uploaded as the new *manuscript* file)

Response: We have copyedited the manuscript ourselves. The errors of grammar and syntax have been corrected in the revised manuscript using Grammarly software. We have named the files as mentioned in the comments.

• Comment 4: We note that the grant information you provided in the ‘Funding Information’ and ‘Financial Disclosure’ sections do not match. 

Response: We have corrected the ‘Funding Information’ and ‘Financial Disclosure’ sections. The grant number is 3/1/3/JRF-2013/HRD-040(10175), dated 10.09.2013. We have mentioned the same in the ‘Funding Information’ section.

• Comment 5: We note that you have indicated that data from this study are available upon request. PLOS only allows data to be available upon request if there are legal or ethical restrictions on sharing data publicly.

Response: We will share the anonymized data with the PLOS and change the data Availability statement.

• Comment 6: Please include captions for your Supporting Information files at the end of your manuscript, and update any in-text citations to match accordingly. Please see our Supporting Information guidelines for more information: http://journals.plos.org/plosone/s/supporting-information.

Response: We have included the captions as mentioned in the comments.

• Comment 7: Please provide additional details regarding participant consent. In the ethics statement in the Methods and online submission information, please ensure that you have specified what type you obtained (for instance, written or verbal, and if verbal, how it was documented and witnessed). If your study included minors, state whether you obtained consent from parents or guardians. If the need for consent was waived by the ethics committee, please include this information.

Response: The study participants were adults, and we took written consent. We explained to the patients regarding the study details. They were also given patient information sheets. After going through them, they provided written permission on the consent form. We have made the necessary changes wherever required.

Comments from Reviewer 1

• Comment 1: a) The Abstract requires a major facelift. 

b) The Methods section simply states "mixed-method" study, without any specific pointers to the respective statistical analysis conducted. 

c) The Results section doesn't state the strength and direction of the findings (via estimates and p-values); this may not be appealing to a reader interested in also understanding the strength of the directions, and findings.

Response 1: 

a) Thank you for pointing these out. We agree with the comments. We have re-written the abstract. In the revised manuscript the new abstract can be found on page number 3 to 4. 

b) The statistical analysis conducted is mentioned in the statistical analysis of the methods section of the manuscript. 

SPSS 23.0 was used to determine count, percentage, mean, SD, median and minimum-maximum for the quantitative data. For the qualitative data, framework analysis was performed. It involved the steps of familiarization, identification of thematic framework, coding, charting and mapping & interpretation. The codes with similar meanings were grouped under subcategories. New themes were allocated appropriately. For the qualitative data Excel spread sheets were used for data management only and codes / themes were entered in spread sheet after developing them on paper (manually).

For the mixed-method analysis the results from the qualitative and quantitative findings were integrated by triangulation during the interpretation stage. Both the findings were side-by-side compared and contrasted to identify the outcomes that agree or converge to build the overall conclusion.

Suggestions have been incorporated on pages numbers 9 and 10.

c) This is a cross-sectional concurrent mixed method triangulation study. For the quantitative data only univariate analysis in form of frequencies, percentage, mean, median, SD and minimum-maximum was performed. The results of these cannot be presented as estimates or p-values. Also, the quantitative aspect is analysed and reported as per the standard procedure of reporting quantitative research and it too cannot be presented as p-values or strength of directions, etc. 

Comment 2: The sample size/power statement does not specifically mentions a desired effect size, the statistical test used, and whether it was computed "using the primary response variable". In fact, it was hard to understand which one is the primary response, and which are secondary (if any).

Response: We missed mentioning the effect size in the calculated sample size. The sample size was calculated using the primary response variable, i.e., PCOS awareness/information status. The precision (corresponding to effect size) was taken as 5%. 

The secondary outcomes were source of obtaining information, health service utilization rate and shift, treatment-lag, decision making in seeking health care and choosing health agency, barriers to seeking treatment, opinions on treatment satisfaction, and expenditure incurred.

The desired information has been incorporated on page number 8 and line 174, and 191-192 of the manuscript.

• Comment 3: More details on exactly what methods (and how, like which software) were used under the banner of "mixed methods" is missing in the Statistical Analysis section. For example, how was the switching calculated in Table 2? I understand triangulation was done, but it may not be very familiar to a reader with little exposure to mixed methods. Note, this is not the same as running a logistic regression.

Response: For the mixed-method analysis, the qualitative and quantitative findings were integrated by triangulation during the interpretation stage. Both the findings were side-by-side compared and contrasted to identify the outcomes that agree or converge to build the overall conclusion.

For the triangulation, different methods were used to measure the same outcome variable (Details given in Response to reviewers.doc file.) These were analysed separately, but at the time of interpretation and pathway development, these were examined for corroboration of results. 

For calculating the switching SPSS software was used. No sophisticated statistical formulae are required (or were used) for calculating the sequential switching between the Health Care Agencies (HCAs). 

We firstly assigned codes for each HCA visited, i.e., 1 for Allopathy government (AG), 2 for Allopathy private (AP), and 4 for indigenous therapies (IT). 

Subtracting the code value of the previous HCA visited from the next HCA (e.g., HCA1 - HCA2, HCA2 - HCA3, and so on) generated new values or codes. The frequency of these codes gave the number of respondents who switched between different HCA. 

The new codes generated for health care agency switching were: No switch = 0, AP to AG = 1, IT to AP = 2, IT to AG =3, AG to AP = (-1), AP to IT = (-2), and AG to IT = (-3).

Comments from Reviewer 2 

• Comment 1: the abstract section of the study is too long. The method section should summarise concisely the method used. Similarly, the result section of the abstract should only summarise the main results. The conclusion should be more focused and reflect the results of the study. Overall, the abstract should be flawless for the readers, not too extended and more focused. 

Response: Thank you for highlighting the flaws in the abstract. These are very valid points. 

We have revised the abstract, methodology, results, and the conclusion section and tried to incorporate the suggested changes to make them concise and focused.

Requisite modifications have been made in the manuscript under each section.

• Comment 2: a)Financial disclosure, please provide grant number or reference. 

b) In the introduction, the author claimed that “Polycystic ovary syndrome (PCOS), a gynaecological morbidity". PCOS is an endocrine condition affecting women of reproductive age; the primary pathology is hormonal disturbances which drives to metabolic, gynaecological and reproductive consequences.

Response: 

a) Thank you for pointing that out. The grant number is 3/1/3/JRF-2013/HRD-040(10175), dated 10.09.2013. We have added the same details in the financial disclosure on page 24, line 478.

b) We accept our mistake in wrongly quoting it as gynaecological morbidity. We have corrected it to ‘endocrine’. The correction is made on page 5, and line 103.

• Comment 3: In the introduction, the author should refer to the recent "international evidence-based guideline for assessing and managing polycystic ovary syndrome 2018". Which highlighted the importance of evaluating the QoL aspect of PCOS.

Response: We have incorporated the mentioned reference in the manuscript. It is added on page 5, and line 116-118.

• Comment 4: The study was designed as a mixed-method triangulation; this method is suitable when combining quantitative and qualitative methods to answer a specific research question. This method leads to a better explanation and link different aspects of a single research question. 

Response: In our study, we tried to triangulate the results of qualitative and quantitative methods to answer questions with a better explanation.

• Comment 5: In the method section, please move reference 16 to Rotterdam criteria instead of the end of the text. 

Response: We have shifted the references as stated by you. Please refer to page 7, and line 164.

• Comment 6: Data collection: this section should be the intervention section which will reflect the questionnaire administered in the study as the author was used an Excel spreadsheet to collect data. For the quantitative data, it is not clear whether the questionnaire used was a validated questionnaire? What was the scale used to capture responses, for instance, the 7-point-Likert scale?. The author has piloted the questionnaire with eight patients; what were the results? Whether any modification to the questionnaire was made as a result of piloting? 

Response: We want to clarify that the data was not collected in spreadsheets. The responses were noted down as field notes and were later translated and entered into Excel spreadsheet.

We designed our questionnaire based on the literature review as there are no validated questionnaires on studying treatment-seeking behaviour in PCOS. The questionnaire was then validated by experts using face and content validation. 

Not all the questions were on a Likert scale. Only a few were based on this scale.

E.g., How effective was the treatment in managing your PCOS signs and symptoms? Effective � Moderately effective� Not effective � N.A.

We realized that to capture the time lag during pilot testing, we had given limited options in days and weeks. However, the patients were taking months before consulting any health care agency. So, in the final questionnaire, we changed the options and asked them to give the time lag as numbers as days/weeks/months. The options were reduced to evaluate treatment effectiveness as patients had confusion understanding between much effective, somewhat effective, low effective, and very low effective options.

We included the treatment expenditure question when the patients complained about the high investigation and treatment cost.

The results of the pilot study (n=8) are given in the Response to reviewers.doc file.

• Comment 7: COREQ guidelines, please use full text (no abbreviation) when it first used. 

Response: Thank you for suggesting that. Consolidated criteria for reporting qualitative research is the full form for COREQ. It has been added to the manuscript. 

We have given the full form with abbreviation on page 8 and line 185.

• Comment 8: The patient's responses were captured in another language which later translated into English. Does this create any risk of bias for the accurateness of the translation? Did the author use approved translation software or translator? It isn't easy to translate word to word from any other language to English. 

Response: We appreciate your concern on this point. Yes, it is not easy to translate word by word; however, this is the routine protocol for any qualitative study to translate from regional language to English. The interaction with almost all the respondents was in Hindi (local language). We translated the text ourselves. 

No, it does not create any bias as it is translated word by word. For accuracy, we back-translated the verbatim.

In qualitative research, the focus is more on ‘listening’. Usually, open-ended questions are asked, and verbatim responses are recorded. These are translated into the English language to generate codes and generating themes. 

The translation in English is required for disseminating the results as Hindi is not a universal language.

• Comment 9: The technical details should be expanded and clarified to ensure that readers understand exactly what the researchers studied. 

Response: These have been clarified in the manuscript.

• Comment 10: The statistical analysis session is too long; this should be short and concise.

Response: We have tried to concise it.

• Comment 11: The author mentioned, "This study objective was an ancillary part of a larger PhD project entitled “Efficacy of probiotic-based dietary and lifestyle regime in the management of PCOS cases", this should be removed from the method section of the manuscript and should be in the conflict of interest section. 

Response: As suggested, we have shifted this to the conflict of interest section. It is mentioned on page 24, and line 480-481.

• Comment 12: The manuscript lacks clear inclusion/exclusion criteria for the participants

Response: Thank you for highlighting that. We have included clear inclusion/exclusion criteria. The criteria are mentioned on page 7, para 3, and line 162-166.

• Comment 13: In the quantitative section of the results, please give exact numbers instead of writing “remaining”.

Response: We have mentioned the exact numbers in the result section of the text.

• Comment 14: What is “viz” mentioned in the manuscript?

Response: The word ‘viz’ means ‘namely’. To avoid confusion we have removed the word in the text. 

• Comment 15: Table 2 is complicated to follow the trends, very complicated for readers to understand precisely what is happening. It should be simplified further. 

Response: We have tried to simplify the table 2 for better understanding.

Please refer to page 13, Table 2.

• Comment 16: Overall, the results section is very long. This should be more concise and focused. 

Response: You have raised an important point here. We have tried our best to make it concise.

• Comment 17: The results and the discussion should be separate sections. 

Response: Thank you for bringing our attention to this relevant point. We have separated the two.

• Comment 18: The discussion section is very repetitive. Please rewrite this section and make it more concise. 

Response: We have rewritten the discussion section.

• Comment 19: Line 447, Figure 1: Pathways to treatment traversed by PCOS patients. This should be removed from the discussion and only refer to the figure within the text, for example (Figure 1). Please remove any bullet points from the text. This section should be in the result section. 

Response: Thank you for focusing on this point. As mentioned in your comment we have made the changes and moved it to the result section.

• Comment 20: The author claimed, “Five major factors emerged from this data, i.e."; this should be in the result section instead of the discussion. 

Response: That is good suggestion. We have shifted it to the result section.

Please refer to page 17, para 2.

• Comment 21: One of the limitations of this study is that the questionnaire was administered in a language other than English, then translated to English. This could lead to a risk of bias in interpreting the script word to word. 

Response: We have covered this comment in comment 8 (please see).

• Comment 22: The conclusion section is very poor; hardly any main results were mentioned. Additionally, this section is very long; it must be short, concise and summarising the main results of the study. 

Response: We have incorporated your suggestions and revised the conclusion. Please refer to page 22-23, and lines 446-456.

• Comment 23: Reference 1, 23, 25, 26,29,32, 33,34, please add DOI. 

Response: We referred to some of the articles and books from the library. We tried to find the DOI numbers for these on the internet but could not find them. We have added the website links for those instead of the DOI to the reference.

• Comment 24: Overall, the language is unclear, making it difficult to follow. I advise the authors work with a writing coach or copyeditor to improve the flow and readability of the text.

Response: Thank you for highlighting this limitation in our manuscript. The lapse is regretted. We have tried improving the flow and readability of the text. Errors of grammar and syntax have been corrected in the revised manuscript using Grammarly software.

• Comment 25: What this study will add to the field?

Response: This study aimed to understand PCOS patients’ treatment-seeking behaviour, experiences, and perspectives. This aspect has not been studied in depth. 

The study's findings helped develop a treatment pathway followed by women with PCOS and the factors that decide the treatment trajectory. It also highlights major issues and barriers they face while seeking treatment. These results can help policymakers provide better quality care and build better coping strategies for managing the condition.

• Comment 26: Reading this study, again and again, I do not know what precisely the main results are. This should be very clear and consistent throughout the text, from the abstract to the conclusion. 

Response: We have revised the whole manuscript from abstract till the conclusion keeping in mind the valuable comments of all the reviews and editor. We have tried to make the text clear, concise, and consistent throughout the manuscript.

---

## [Decision Letter · Decision Letter 1]

26 Jul 2021

Treatment pathways traversed by polycystic ovary syndrome (PCOS) patients: A mixed-method study

PONE-D-21-07635R1

Dear Dr. Singh,

We’re pleased to inform you that your manuscript has been judged scientifically suitable for publication and will be formally accepted for publication once it meets all outstanding technical requirements.

Kind regards,

Stephen L Atkin, MD

Academic Editor

PLOS ONE

Additional Editor Comments (optional):

Reviewers' comments:

Reviewer's Responses to Questions

**Comments to the Author**

1. If the authors have adequately addressed your comments raised in a previous round of review and you feel that this manuscript is now acceptable for publication, you may indicate that here to bypass the “Comments to the Author” section, enter your conflict of interest statement in the “Confidential to Editor” section, and submit your "Accept" recommendation.

Reviewer #1: All comments have been addressed

Reviewer #2: All comments have been addressed

2. Is the manuscript technically sound, and do the data support the conclusions?

Reviewer #1: (No Response)

Reviewer #2: Yes

3. Has the statistical analysis been performed appropriately and rigorously? 

Reviewer #1: (No Response)

Reviewer #2: I Don't Know

4. Have the authors made all data underlying the findings in their manuscript fully available?

Reviewer #1: (No Response)

Reviewer #2: Yes

5. Is the manuscript presented in an intelligible fashion and written in standard English?

Reviewer #1: (No Response)

Reviewer #2: Yes

6. Review Comments to the Author

Reviewer #1: (No Response)

Reviewer #2: The authors have addressed the comments raised by the reviewer. The readability of the manuscript has improved.

7. PLOS authors have the option to publish the peer review history of their article (what does this mean?). If published, this will include your full peer review and any attached files.

Reviewer #1: No

Reviewer #2: No

---

## [Editor Report · Acceptance letter]

30 Jul 2021

PONE-D-21-07635R1 

Treatment pathways traversed by polycystic ovary syndrome (PCOS) patients: A mixed-method study 

Dear Dr. Singh:

I'm pleased to inform you that your manuscript has been deemed suitable for publication in PLOS ONE. Congratulations! Your manuscript is now with our production department. 

Kind regards, 

on behalf of

Dr. Stephen L Atkin 

Academic Editor

PLOS ONE